# Conventional and Green Rubber Plasticizers Classified through Nile Red [E(NR)] and Reichardt's Polarity Scale [E$_T$(30)]

Franco Cataldo 

Actinium Chemical Research Institute, Via Casilina 1626A, 00133 Rome, Italy; franco.cataldo@fastwebnet.it

**Abstract:** After a survey on polymer plasticization theories and conventional criteria to evaluate polymer–plasticizer compatibility through the solubility parameter, an attempt to create a polymer–plasticizer polarity scale through solvatochromic dyes has been made. Since Reichardt's E$_T$(30) dye is insoluble in rubber hydrocarbon polymers like polyisoprene, polybutadiene and styrene–butadiene copolymers and is not useful for the evaluation of the hydrocarbons and ester plasticizers, the Nile Red solvatochromic dye was instead used extensively and successfully for this class of compounds. A total of 53 different compounds were evaluated with the Nile Red dye and wherever possible also with Reichardt's E$_T$(33) dye. A very good correlation was then found between the Nile Red scale E(NR) and Reichardt's E$_T$(30) scale for this class of compounds focusing on diene rubbers and their typical hydrocarbons and new ester plasticizers. Furthermore, the E(NR) scale also shows a reasonable correlation with the total solubility parameter calculated according to the Van Krevelen method. Based on the above results, some conclusion was made about the compatibility between the diene rubbers and the conventional plasticizers, as well as a new and green plasticizer proposed for the rubber compounds.

**Keywords:** rubbers; plasticizers; solvatochromic dyes; Nile Red dye; Reichardt's dye E$_T$(30); Reichardt's dye E$_T$(33); compatibility; solubility parameter



## 1. Introduction

Plasticizers are liquids at ambient temperature with a relatively high molecular weight or, less frequently, low-melting-point solids, which are added to a polymer matrix to change its viscoelastic properties [1–9]. The changes in the physical properties of the polymer matrix have many consequences starting from an improved processability, passing through an increased flexibility of the resulting polymer compound, and leading to improved low-temperature performances [1–9]. The latter result is achieved because the plasticizer is often a liquid characterized by a lower glass transition temperature (T$_g$) than the guest polymer matrix, leading to a shift of the compound T$_g$ toward lower temperatures. Indeed, the plasticizer efficiency is often measured by the degree of glass transition temperature shift toward lower temperatures of the resulting plasticized compound (T$_g^c$) with respect to the glass transition of the raw polymer (T$_g^P$) so that $\Delta T_g = T_g^P - T_g^c$ [1–9]. However, the $\Delta T_g$ is most pronounced in polymers with rigid chains (e.g., PVC), whereas the plasticizer causes a shift of the order of $100-160\ °C$. On the other hand, the effect of plasticizers on already flexible and rubber-like polymers is much less pronounced and limited to a $\Delta T_g$ shift to just a few tens of $°C$ toward lower temperatures [1–9]. For rubbers already characterized by low glass transition temperature values (e.g., natural rubber T$_g$ = $-72\ °C$ or high cis-polybutadiene with Tg = $-105\ °C$), the plasticizer effect can cause a $\Delta T_g \approx -10\ °C$ [10,11]. For each given type of polymer or polymer blend, the efficiency of a plasticizer is measured by the degree of $\Delta T_g$ it is able to cause with respect to another plasticizer. Furthermore, at least for polar polymers (and with a series of limitations), a simple relationship has been found: $\Delta T_g = kn$ with k being a proportionality constant and n being the moles of plasticizer added, suggesting that the glass transition temperature shift is directly proportional to

the amount of the plasticizer added [3]. On the other hand, for apolar polymers, it holds a similar relationship: $\Delta T_g = k'\varphi$, where $\varphi$ is the volume fraction of the plasticizer [3]. Another simple relationship for the determination of the compound glass transition is the following [8]:

$$1/T_g^c = \omega_1/T_1 + \omega_2/T_2 \tag{1}$$

where $\omega_1$ and $\omega_2$ are the weight fractions of polymer and plasticizer, respectively, while $T_1$ and $T_2$ values are the respective glass transition temperatures. Other relationships for the estimation of the glass transition of a plasticized compound can be found in ref. [8].

At this point, it is worth very shortly surveying the three main theories of plasticization mechanisms: the lubricity theory, the gel theory and the free volume theory. It is also interesting to note that the latter theory also includes the other two [1–9].

The lubricity theory [1–9] starts from the observation that in a pure polymer, the resistance to deformation and flow derives from the intermolecular friction between adjacent polymer chains which are in direct contact. The introduction of the plasticizer molecules between the polymer chains facilitates the movement of the chain segments through a slippage mechanism provided by the lubricating action of the plasticizer.

The gel theory [1–9] represents a further step ahead with respect to the lubricity theory. It starts from the idea that in amorphous polymers, the resistance to deformation derives from a model structure of the polymer intended to be a three-dimensional or honeycomb-type structure, which can also be defined as a gel structure. Such a gel structure is conceived as derived from loose attachments or secondary forces that occur at rather regular intervals along the polymer chains and hence in the matrix. The introduction of a plasticizer in such honeycomb structures has the effect of breaking the loose attachments and masking the center of forces by preventing the reformation of the three-dimensional macromolecular interaction. It is admitted that the masking action of the force centers derives from the fact that the polymer chain segments are solvated by the plasticizer. Solvation is always intended in a dynamic way so that each chain segment is solvated and de-solvated and, sometimes, the masking action is lost for a while. This implies an increased flexibility and flowability of the polymer chain and the resulting plasticized compound. Naturally, the chemical nature of each plasticizer exerts a different effect on the masking effect of the secondary force centers, leading to the individual effect of each plasticizer.

The free volume theory [1–9] starts from the observation that the free volume in a polymer is a measure of the internal available space for the motion of the chain segments, the chain ends and the side groups. The free volume reaches the minimum value, say 2.5% of the volume of the given polymer body, when it is cooled below its glass transition temperature. Indeed, below the Tg, because of the limited free volume available, the mentioned chain, end and side group movements are frozen and the polymer is in a glassy state. An increase in the motion of these moieties of the polymer can be achieved by (1) heating, (2) the addition of a plasticizer (which, being a low-molecular-weight molecule, increases the number of chain ends dramatically), (3) introducing branching or bulky side groups to the main polymer chain, and (4) inserting more flexible chain segments into the main polymer chain. Thus, any action that leads to an increase in the free volume of a polymer is measurable by a shift of the glass transition toward lower temperatures. The simplest action, which does not cause a chemical modification of the polymer, is just the physical addition of a plasticizer. This is known as external plasticization to distinguish it from internal plasticization, which instead is due to the chemical modification of the polymer. It is evident that the free volume theory is the most convincing and includes all the notions of the lubricity and gel theories. In fact, the plasticizer fills the available free volume and creates extra free volume; the first molecular layer of the plasticizer is adsorbed on the polymer chain segments and provides a certain degree of solvation, shielding the force centers of interaction between chains and preventing polymer networking reformation on cooling. The excess plasticizer molecules, not interacting directly with the polymer chain segments, act as a volume filler of the free volume created by the first layer of molecules solvating the chain segments and may provide a lubricating effect under deformation and

flow. Additional discussion about the free volume theory and recent developments can be found in ref. [8].

The key practical problem in the use of plasticizers is the evaluation of the compatibility between the polymer, the polymer blend and the plasticizers. A general and updated survey can be found in ref. [9]. However, the most popular and accessible approach to evaluating the polymer–plasticizer compatibility involves the use of the solubility parameter either in rubber compounds [12] or in plastics [13].

The solubility parameter has been defined by Hildebrand and Scott [12] as follows:

$$\delta = [(\Delta H_{vap} - RT)/V_m]^{0.5} \tag{2}$$

The evaporation enthalpy $\Delta H_{vap}$ was taken as the parameter of the cohesion energy between molecules minus the thermal energy needed to separate them (RT) divided by the molar volume $V_m$. Equation (2) can be re-written as

$$\delta = [(E_{coh})/V_m]^{0.5} \tag{3}$$

The cohesive energy $E_{coh}$ of a substance in a condensed state is defined as the increase in internal energy $\Delta U$ per mole of substance if all the intermolecular forces are eliminated.

Hansen [13] showed that the solubility parameter proposed by Hildebrand and Scott does not take into account the contribution of polar forces and hydrogen bonding; therefore, a more complex solubility parameter has been proposed:

$$\delta^2 = \delta_d{}^2 + \delta_p{}^2 + \delta_h{}^2 \tag{4}$$

derived from the contribution of three components of the cohesive energy:

$$E_{coh} = E_d + E_p + E_h \tag{5}$$

which are due to the contribution of dispersion and polar forces plus a hydrogen bonding contribution, respectively.

It is possible to calculate the solubility parameters and the solubility parameter components of almost all molecules and polymers by a group contribution method [14]. For this purpose, as explained by Van Krevelen [14], it is useful to introduce the molar attraction constant simply defined as

$$\varphi = (E_{coh} \, V_m)^{0.5} \tag{6}$$

A set of equations was proposed by Van Krevelen [14] for the calculation of the solubility parameter components using molar attraction by a group contribution methodology:

$$\delta_d = (\sum \varphi_d)/V_m \tag{7}$$

$$\delta_p = (\sum \varphi_p{}^2)^{0.5}/V_m \tag{8}$$

$$\delta_h = [(\sum E_h)/V_m]^{0.5} \tag{9}$$

The total solubility parameter can be calculated as follows:

$$\delta_t = (\delta_d{}^2 + \delta_p{}^2 + \delta_h{}^2)^{0.5} \tag{10}$$

It can be observed from Equation (9) that the hydrogen bond parameter $\delta_h$ cannot be calculated from the molar attraction, but directly from the hydrogen bonding energy $E_h$ [14]. There are numerous ways of evaluating the solubility of a given polymer P in a given solvent S; Van Krevelen [14] suggests the criteria imposed by the following equation:

$$\Delta\delta = [(\delta_{d,P} - \delta_{d,S})^2 + (\delta_{p,P} - \delta_{p,S})^2 + (\delta_{h,P} - \delta_{h,S})^2]^{0.5} \tag{11}$$

To predict solubility,

$$\Delta\delta \leq 5 \tag{12}$$

Alternatively, Hansen [13] proposed a more sophisticated and relatively complex approach for the evaluation of the solubility of a polymer in a solvent.

A simpler and more practical approach regards the direct adoption of the total solubility parameter $\delta_t$, eventually determined according to Equation (10), to evaluate the solubility between a polymer and a plasticizer or a solvent, which conforms to the criteria imposed by the following equation:

$$|\Delta\delta| = (\delta_{t,P} - \delta_{t,S}) \leq 3.0 \tag{13}$$

a criterion proposed by Brydson [12].

By using the Van Krevelen methodology [14] in previous works, we have calculated the solubility parameters of fullerenes [15] and their solubility in fatty acid esters and glycerides [16]. Similarly, it was through the calculated solubility parameters according to the Van Krevelen methodology [14] that the compatibility between biodiesel and diene rubbers as well as other typical petroleum-derived plasticizers used in rubber compounding was calculated [17].

In the present work, we wish to show an alternative or complementary approach to the solubility parameter to evaluate in a practical way the compatibility between a plasticizer and a polymer matrix (in particular a rubber polymer) by also classifying the plasticizers and the rubbers through Reichardt's polarity scale $E_T(30)$ or through a complementary scale. After all, Reichardt's polarity scale was also successfully applied for the first time in the selection of a bonding agent for a rocket propellant composite [18].

## 2. Materials and Methods

### 2.1. Materials

All plasticizers, solvents and polymers, unless otherwise stated, were purchased from Merck-Aldrich (Darmstadt, Germany– St.Louis, MO, USA). The petroleum-based plasticizers or extenders used in the rubber (tire) industry were commercially available products sourced from the market of rubber chemicals and additives. Namely, these plasticizers were T-DAE (treated distillate aromatic extract) and MES (mild extract solvate). Only the product Nytex BIO 6200 (a blend of naphthenic oil and fatty acids from renewable sources) was obtained from Nynas (Stockholm, Sweden). The methyl ester of rapeseed oil was a commercial biodiesel we used as a rubber compound plasticizer in our earlier work [17]. Another commercially available plasticizer of this study was the methyl ester of coconut oil. These methyl esters are prepared on an industrial scale by transesterification of the corresponding glycerides with methanol. The sunflower oil (high oleic content) and the soybean oil of the present work were industrial products obtained from the market of rubber chemicals. There is a trend in the patent [19,20] and in the open literature, e.g., [21], to adopt vegetable oils in tire treads as a substitute for petroleum-based plasticizers to produce greener tires. The most interesting vegetable oils appear to be soybean [19] and sunflower oil [20], also including a combination of both oils [21] as plasticizers in tire treads. Regarding the rubber samples, cis-1,4-polybutadiene (Europrene Neocis 60) was obtained from Versalis (Ravenna, Italy), synthetic cis-1,4-polyisoprene sample was SKI-5PM produced by JSC Sterlitamak Petrochemical Plant (Russia) and solution styrene–butadiene rubber (S-SBR) was obtained from Arlanxeo (Dormagen, Germany). The S-SBR was FX5000 grade, characterized by 50% vinyl content and 20% styrene content. Epoxidized natural rubber (ENR-25) was supplied by Ekoprena (Kuala Lumpur, Malaysia). The nitrile rubber sample was sourced from Versalis (Ravenna, Italy), and it was the Europrene N3345 with 33% acrylonitrile content. All the other polymers including the liquid polymer samples were purchased from Merck-Aldrich (Darmstadt, Germany– St.Louis, MO, USA).

The solvatochromic dyes E$_T$(30) and Nile Red, whose chemical structures are shown in Scheme 1, were purchased from Merck-Aldrich (Germany–USA). The solvatochromic dye E$_T$(33) was obtained from Fluka (Buchs, Switzerland).

2,6-diphenyl-4-(2,4,6-triphenyl-1-pyridinio)phenolate
E$_T$(30) dye

2,6-dichloro-4-(2,4,6-triphenyl-1-pyridinio)phenolate
E$_T$(33) dye

9-diethylamino-5H-benzo[$\alpha$]phenoxazinone
Nile Red dye

**Scheme 1.** The solvatochromic dyes used in the present work.

*2.2. Determination of the Maximum Absorbance with the Solvatochromic Dyes in Liquid Samples*

For all plasticizers, solvents and liquid polymers, the dissolution of the minimal quantity of the selected solvatochromic dye was made in a beaker by stirring. For viscous liquids, gentle heating was applied to accelerate the dissolution of the dye. The typical volume of each sample under analysis was 10 mL and the dye added was much less than a spatula tip. Before making any spectrophotometric measurement on our Shimadzu UV2450 equipped with thermostated cells kept at 25 °C, a back correction was made with quartz cuvettes filled with the pure liquid sample under study. This operation is crucial, especially with yellow or even brown liquid samples, for instance, the plasticizers used in the tire industry, to reduce or minimize the color interference of the sample. After the back-correction step, the electronic absorption spectrum of the solvatochromic dye in the selected liquid was recorded using the reference cuvette filled with the reference liquid without any dye.

*2.3. Determination of the Maximum Absorbance with the Solvatochromic Dyes in Solid Thin Films of Polymer Sample*

Regarding the polymer thin films with the solvatochromic dye (typically Nile Red) embedded inside, the sample preparation involved the following steps. The selected polymer or rubber cut into the smallest possible pieces was weighed in a flask (typically 500–600 mg) and dissolved in about 13 g of dichloromethane (CH$_2$Cl$_2$). Only for epoxidized

natural rubber (ENR-25) and for nitrile rubber (Europrene N3345) was it necessary to use tetrahydrofuran (THF) as a solvent instead of $CH_2Cl_2$. Once the polymer was fully dissolved, less than a tip of a spatula of Nile Red was added to the solution and stirred till a complete dissolution of the dye was achieved. The resulting homogeneous solution was poured into an optical glass dish with a diameter of 11 cm and the solvent was allowed to evaporate slowly under a fume hood. After the complete evaporation of the solvent, the resulting dyed polymer film in the dish was heated to 60–70 °C to permit the complete evaporation of any solvent trace. After cooling, the electronic absorption spectrum of the dyed polymer film was directly collected through the optical glass dish. Only in a couple of cases, i.e., with polystyrene and poly(lactic acid), was it possible to separate the polymer film from the glass dish as a free-standing thin solid film. In the latter two cases, the electronic absorption spectra were recorded directly on the free-standing thin solid films.

It is known from the literature [22] that the solvatochromic dye may separate into the form of microcrystals on the polymer surface leading to non-useful absorption maxima readings. This potentially undesired effect was minimized by using the minimum possible amount of dye during the preparation of the composite. Furthermore, Nile Red dye is characterized by a high solubility in hydrocarbon polymers (in contrast to the $E_T(30)$ and $E_T(33)$ dyes). Consequently, the potential blooming was therefore completely suppressed.

The potential solvent effect was checked in the case of polybutadiene. This polymer was dissolved both in $CH_2Cl_2$ and in THF. The resulting two different polymer films with Nile Red dye prepared on glass dishes were analyzed spectrophotometrically, obtaining the same maximum absorption value. Of course, the final "drying" step of the film under moderate heating is crucial to remove all the solvent traces, which otherwise may affect the position of the maximum absorption.

## 3. Results

### 3.1. General Aspects of Nile Red Solvatochromic Dye with Respect to $E_T(30)$ Dye

The challenge of this work is to evaluate, through a solvatochromic scale, the potential compatibility between polymers and plasticizers with special attention to rubber polymers and plasticizers used in the rubber and tire industry. This new approach could be a complementary and experimental way with respect to the evaluation and estimation of the solubility parameter of each component as conducted, for example, in a previous work [17] using the Van Krevelen methodology [14]. The key problem faced by the present work is represented by the fact that the fundamental Reichardt's dye, $E_T(30)$ of Scheme 1, is insoluble in hydrocarbons, and it is also not suitable for ester plasticizers either because of the relatively low solubility but also because the small residual acidity in the esters causes the protonation of the phenolate oxygen anion, leading to the disappearance of the long-wavelength solvatochromic charge transfer (CT) band [23]. Indeed, the $E_T(30)$ values of hydrocarbons were all determined through another complementary penta-*t*-butyl-substituted betaine dye called $E_T(45)$ [24], which is not easily accessible and furthermore, it is exactly as sensitive to protonation as the $E_T(30)$ dye. Thus, the Reichardt dye suitable for polar media with weak acidity is the $E_T(33)$ dye (see Scheme 1) [23], which in fact was adopted in the present work. The symmetrical chlorine substitution in the ortho-position to the phenolate group changes the pKa of the conjugated acid of the dye $E_T(33)$ by four orders of magnitude with respect to $E_T(30)$, making the former dye suitable for weakly acidic media [23]. Since the $E_T(30)$ but also the $E_T(33)$ values expressed in kcal/mol are both determined through the well-known equation [23,24]

$$E_T = 28{,}591\ \lambda^{-1} \tag{14}$$

where $\lambda$ is the maximum absorption of the long-wavelength CT band of the pyridinium-N-phenolate betaine dye in a given liquid medium; the conversion from $E_T(33)$ values to $E_T(30)$ can be achieved through the following equation [23]:

$$E_T(30) = 0.9953\ E_T(33) - 8.1132 \tag{15}$$

Despite the availability of $E_T(33)$, the potential measurement of the polarity in certain hydrocarbon-based plasticizers, in certain apolar polymers and in certain esters remains unaffordable.

In a seminal paper belonging to this Special Issue of "*Liquids*", Acree and Lang [25] have shown an interesting and sophisticated approach that permits the estimation of the $E_T(30)$ values of certain "difficult" substrates where the direct polarity measurement is hindered for a series of reasons, as in the present case for apolar polymers (e.g., rubbers) and certain plasticizers.

An alternative to the above approach is to resume the solvatochromic dye known as Nile Red. The chemical structure of Nile Red is shown in Scheme 1 and the full chemical name of the dye is 9-diethylamino-5H-benzo[α]phenoxazinone [26]. It is a very stable fluorescent dye used for staining biological tissues, being highly lipophilic [27]. Certain structural analogies between Nile Red and phenol blue have been known for a long time, including the interesting solvatochromic behavior of the former dye [28]. Indeed, Nile Red is not only soluble in hydrocarbons but also in fats, where $E_T(30)$ is insoluble. In acid media, Nile Red does not lose its solvatochromic behavior, in contrast with the $E_T(30)$ dye [28]. In these instances, Nile Red is completely complementary to $E_T(30)$ and it is the ideal dye for the present work, where the substrates are apolar polymers (e.g., rubbers), hydrocarbons and ester plasticizers, but also triglycerides.

Deye, Berger and Anderson [28,29] have already performed a systematic study of Nile Red in comparison to $E_T(30)$ dye, showing that with the former dye, it is possible to measure the solvent polarity in many more liquids than those directly accessible by the $E_T(30)$, including fluorinated molecules and supercritical $CO_2$. As in the case of $E_T(30)$, the electronic absorption maximum of Nile Red can be expressed in kcal/mol according to Equation (14), so that an E(NR), i.e., Nile Red, scale for solvent can be constructed. A recognized disadvantage of the E(NR) scale with respect to the $E_T(30)$ scale regards the fact that the band shift of the Nile Red dye in media of different polarities is less pronounced than the case of the band shift offered by Reichardt's dye [28]. This implies an intrinsic lower sensitivity of the E(NR) to polarity change with respect to the $E_T(30)$. Furthermore, as can be seen in Figures 1 and 2, the absorption maxima of Nile Red and Reichardt's dyes vary in the opposite direction to the polarity of the medium, i.e., Nile Red is a positively solvatochromic dye, whereas Reichardt's dye is a negatively solvatochromic probe [30].

Figure 1 shows that the correlation between E(NR) and $E_T(30)$ is not linear but quite complex when considering all data available for any type of solvent (excluding the fluorinated molecules):

$$E(NR) = -0.0012\,[E_T(30)]^3 + 0.1728\,[E_T(30)]^2 - 8.1534\,[E_T(30)] + 182.17 \qquad (16)$$

with a correlation coefficient $R^2 = 0.888$.

However, when considering only certain homogenous classes of solvents, for example, non-HBD solvents, simpler and more linear correlations are obtained, as will be shown later.

It is also interesting to note that the attention toward the Nile Red dye as a solvatochromic probe alternative or complementary to $E_T(30)$ is steady from a theoretical point of view but also from a practical point of view. The theoretical analysis focuses on the nature of the charge transfer transition of Nile Red, which makes it an effective probe. A twisted intramolecular charge transfer transition was advocated for this dye [31–37], and the ground and excited dipole moments of the dye were also estimated. The Nile Red dye is steadily employed in many measurements including probing hydrocarbon liquids [36], absorption and fluorescence in a series of alcohols [37], analysis of green chemistry solvents' polarity [38], study of anisotropic liquids [39], in zeolites [40], detection of lipid order heterogeneity in cells [41], and detection of hydrogen bonding strength in microenvironments [42], limiting the numerous fields of current applications.

**Table 1.** Measured or calculated ET(30) and E(NR) values of plasticizers, rubber and polymers.

| PLASTICIZER, SOLVENT OR POLYMER | $E_T(30)$ Kcal/mol | $E(NR)$ Kcal/mol | References or Notes on the $E_T(30)$ Values | References or Notes on the $E(NR)$ Values | Solubility Parameter $\delta_t$ in (MPa)$^{0.5}$ Calc. According to Ref. [14] |
|---|---|---|---|---|---|
| Isooctane | **30.7** | **58.77** | ref. [24] | this work | 14.6 |
| N-hexane and cyclohexane | **31.0** | **58.75** | ref. [24] | this work | 15.1 |
| Tetradecane | **31.0** | **58.21** | ref. [24] | this work | 16.2 |
| **SQUALANE** | **31.0** | **58.12** | estimated | this work | 15.6 |
| Decalin | **31.2** | **57.87** | ref. [24] | this work | 16.9 |
| **SQUALENE** | **36.5** | **57.64** | this work fm $E_T(33)$ | this work | 16.0 |
| **Liquid cis-POLYISOPRENE (Liq-IR)** | **35.4** | **57.55** | this work fm $E_T(33)$ | this work | 16.6 |
| **Liquid 1,2-POLYBUTADIENE** | *33.0* | **57.32** | calculated (*) | this work | 17.1 |
| **Liquid cis-POLYBUTADIENE (Liq-BR)** | **33.2** | **57.25** | calculated (*) | this work | 17.1 |
| Cyclohexene | **32.2** | **57.19** | ref. [24] | this work | 15.4 |
| **POLYBUTADIENE thin film (BR)** | *34.2* | **56.96** | calculated (*) | this work | 17.1 |
| **POLYISOPRENE thin film (IR)** | *34.6* | **56.82** | calculated (*) | this work | 16.6 |
| Oleyl Oleate | **36.2** | **56.27** | this work fm $E_T(33)$ | this work | 16.3 |
| **S-SBR with styrene 21% & vinyl 50%** | *36.8* | **56.17** | calculated (*) | this work | 17.4 |
| **T-DAE oil** | **36.8** | *56.2* | this work fm $E_T(33)$ | calculated (**) | 17.3 |
| **POLYSTYRENE thin film** | *37.8* | **55.84** | calculated (*) | this work | 17.5 |
| Di-n-butyl PHTHALATE | **39.5** | *55.3* | ref. [24] | calculated (**) | 17.8 |
| Dimethyl PHTHALATE | **40.7** | *55.0* | ref. [24] | calculated (**) | 18.0 |
| Diisododecyl ADIPATE | **36.4** | **54.98** | this work fm $E_T(33)$ | this work | 17.5 |
| Ethyl OLEATE | **40.5** | **54.85** | this work fm $E_T(30)$ | this work | 17.3 |
| **MES oil** | **37.3** | **54.83** | this work fm $E_T(33)$ | this work | 16.7 |
| **Biodiesel fm rapeseed oil** | *41.4* | **54.77** | calculated (*) | this work | 16.6 |

**Table 1.** *Cont.*

| PLASTICIZER, SOLVENT OR POLYMER | $E_T(30)$ Kcal/mol | $E(NR)$ Kcal/mol | References or Notes on the $E_T(30)$ Values | References or Notes on the $E(NR)$ Values | Solubility Parameter $\delta_t$ in (MPa)$^{0.5}$ Calc. According to Ref. [14] |
|---|---|---|---|---|---|
| Diethylhexyl SEBACATE | 36.3 | 54.76 | this work fm $E_T(33)$ | this work | 17.0 |
| Coconut methyl ester | 41.6 | 54.70 | calculated (*) | this work | 16.7 |
| Dimethyl SEBACATE | 41.8 | 54.7 | this work fm $E_T(33)$ | calculated (**) | 18.1 |
| Diethylhexyl ADIPATE (DOA) | 36.4 | 54.67 | calculated (*) | this work | 17.8 |
| Dibutyl SEBACATE | 42.3 | 54.5 | this work fm $E_T(33)$ | calculated (**) | 17.8 |
| **Soybean oil** | 42.1 | 54.55 | calculated (*) | this work | 17.2 |
| **Nytex BIO 6200** | 42.1 | 54.55 | calculated (*) | this work | 16.7 |
| **Epoxidized natural rubber (ENR-25)** | 42.9 | 54.32 | calculated (*) | this work | 17.2 |
| Dioctylterephthalate (DOTP) | 43.0 | 54.27 | calculated (*) | this work | 17.4 |
| **Sunflower oil (high oleic content)** | 43.5 | 54.13 | calculated (*) | this work | 16.7 |
| Ethyl PALMITATE | 43.9 | 54.0 | this work fm $E_T(30)$ | calculated (**) | 16.8 |
| Diethyl AZELATE | 45.1 | 54.08 | this work fm $E_T(33)$ | this work | 17.3 |
| Methyl undecenoate | 44.0 | 53.99 | calculated (*) | this work | 17.9 |
| PEG dioleate | 44.6 | 53.79 | calculated (*) | this work | 17.7 |
| PEG monooleate | 45.2 | 53.60 | calculated (*) | this work | 17.6 |
| **Polymethylmethacrylate film (PMMA)** | 45.8 | 53.44 | calculated (*) | ref. [43] | |
| **Poly(Lactic acid) film (PLLA)** | 45.8 | 53.44 | calculated (*) | this work | 19.5 |
| Bis(THFA) ADIPATE | 46.2 | 53.30 | this work fm $E_T(33)$ | calculated (**) | 19.0 |
| Ethyl levulinate | 46.2 | 53.30 | calculated (*) | this work | 18.9 |
| Dioctylphthalate (ethylhexyl) (DOP) | 46.4 | 53.24 | calculated (*) | this work | 18.2 |
| **Nitrile Rubber with 33% ACN film** | 47.0 | 53.07 | calculated (*) | this work | 19.4 |
| Bis(THFA) SEBACATE | 47.1 | 53.0 | this work fm $E_T(33)$ | calculated (**) | 18.4 |
| Bis(THFA) AZELATE | 47.3 | 53.0 | this work fm $E_T(33)$ | calculated (**) | 18.6 |

**Table 1.** *Cont.*

| PLASTICIZER, SOLVENT OR POLYMER | $E_T(30)$ Kcal/mol | E(NR) Kcal/mol | References or Notes on the $E_T(30)$ Values | References or Notes on the E(NR) Values | Solubility Parameter $\delta_t$ in $(MPa)^{0.5}$ Calc. According to Ref. [14] |
|---|---|---|---|---|---|
| THFA OLEATE | 47.4 | 53.0 | this work fm $E_T(33)$ | calculated (**) | 17.8 |
| THFA LAURATE (30 °C) | 48.0 | 52.8 | this work fm $E_T(33)$ | calculated (**) | 17.2 |
| THFA PELARGONATE (30 °C) | 49.3 | 52.4 | this work fm $E_T(33)$ | calculated (**) | 18.9 |
| Tetrahydrofurfuryl alcohol | 49.9 | 52.2 | this work fm $E_T(30)$ | calculated (**) | |
| L-(-) ethyl lactate | 51.1 | 52.16 | ref. [24] | this work | |
| **Polyethylene glycol (PEG-400)** | 49.7 | 52.15 | this work fm $E_T(30)$ | this work | 19.7 |
| **Polytetrahydrofuran (PolyTHF)low Mw** | 49.3 | 52.12 | this work fm $E_T(30)$ | this work | 18.9 |
| Water | 63.1 | 48.21 | ref. [24] | this work | |

(*) Calculated with E(NR) = $-0.303\ E_T(30) + 67.31$; (**) Calculated with $E_T(30) = [E(NR)—67.31]/-0.303$; THFA = Tetrahydrofurfuryl.

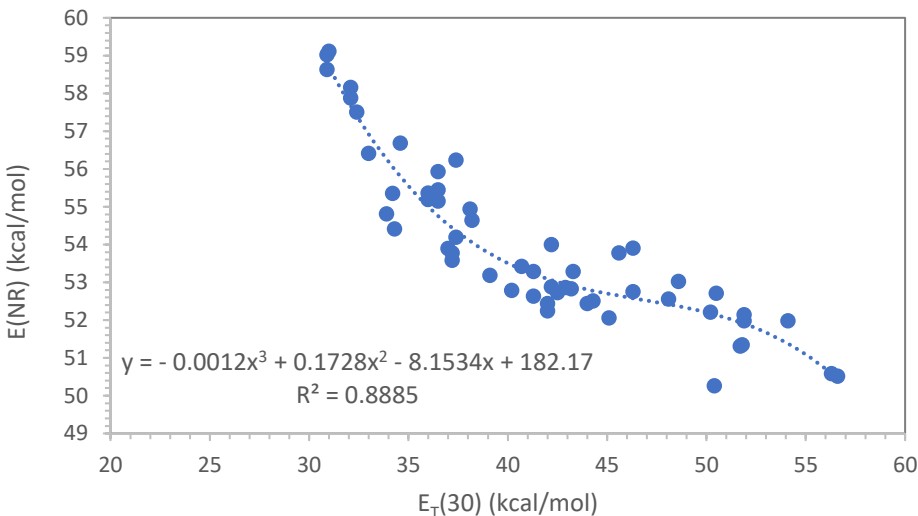

**Figure 1.** Correlation between E(NR) and $E_T(30)$ using the data of ref. [28] and excluding the fluorinated molecules.

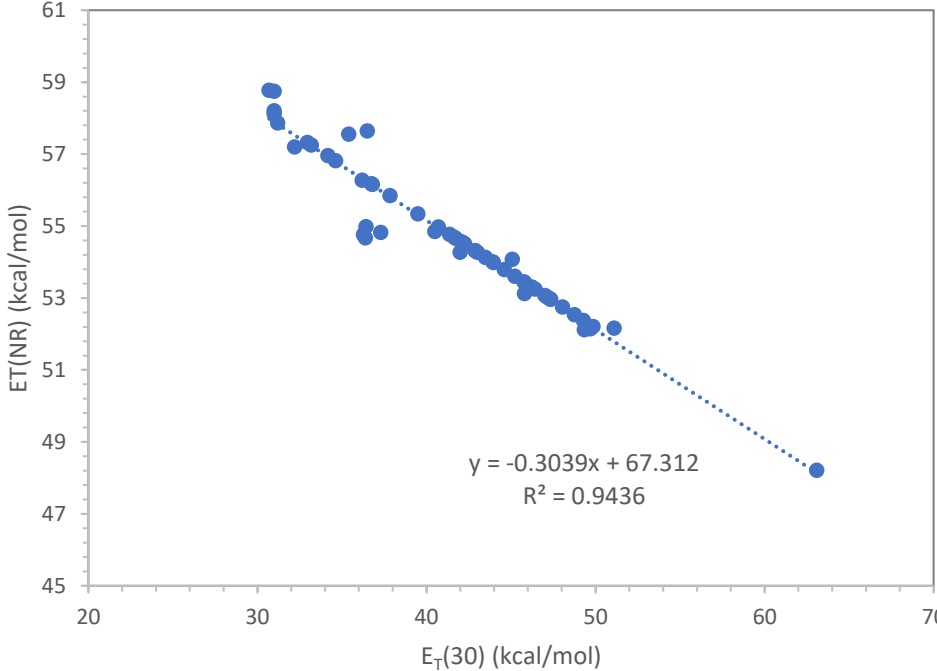

**Figure 2.** Correlation between E(NR) and $E_T(30)$ using all the experimental data of Table 1.

### 3.2. Determination of Rubber and Polymer Polarity with Nile Red Solvatochromic Dye

As detailed in Section 2.3, the micropolarity of the rubber polymers was determined with Nile Red solvatochromic dye either using model compounds or liquid polymers where the dye was soluble or using thin solid-state polymer films where the dye was embedded in the polymer matrix with the aid of a solvent, which was then removed completely. The basic idea of the latter approach derives from the seminal work of Dutta et al. [43], who used Nile Red as a micropolarity probe for plastics in polymethylmethacrylate (PMMA) and polyvinylalcohol films. We also extended the same approach to rubber polymers like cis-1,4-polyisoprene, cis-1,4-polybutadiene, the styrene–butadiene copolymer, epoxidized natural rubber and nitrile rubber. The micropolarity of polystyrene and poly(lactic acid) films was also studied, whereas the E(NR) value reported by Dutta et al. [43] was adopted for PMMA.

Table 1 reports the experimental E(NR) values for all the above-mentioned rubbers and other liquids. In Table 1, the apolar compounds are at the top of the table and the most polar at the bottom of it.

Regarding the rubber and other polymers studied, there is a logical trend from apolar to polar compounds starting with the liquid model compounds squalane and squalene (squalane is the liquid analog of the ethylene–propylene copolymer, while squalene can be considered a liquid model compound of natural rubber or cis-1,4-polyisoprene as discussed elsewhere [44]) as the most apolar, followed by the liquid polyisoprene and polybutadiene and then by the thin solid films of polybutadiene (BR) and polyisoprene (IR). The copolymer styrene–butadiene rubber (S-SBR) with 21% styrene content is found to be more polar than either BR or IR as expected. S-SBR shows E(NR) values in the same range as polystyrene thin solid film and the petroleum-based rubber plasticizer T-DAE, whose aromatic content is in the range of 20–25%.

As expected, polar polymers are appropriately positioned in the E(NR) scale, with epoxidized natural rubber (ENR-25) showing an E(NR) value of 54.32 kcal/mol with respect to 57.55 kcal/mol of liquid IR and 56.82 kcal/mol of the solid film of IR.

Even more polar than ENR-25 are polymethylmethacrylate (PMMA) and poly(lactic acid) (PLLA), both with E(NR) = 53.44 kcal/mol, followed by nitrile rubber (NBR) with 33% acrylonitrile (ACN) content and a Nile Red electronic transition at 53.07 kcal/mol. The most polar polymers (among those studied in the present work) were two oligomers: polyethylene glycol (PEG-400) and polytetrahydrofuran liquid oligomer (polyTHF), both at 52.1 kcal/mol. Being oligomers, the latter two compounds have their micropolarity affected by the OH end groups which, of course, play an important role in affecting their polarity.

### 3.3. Determination of Rubber Plasticizer Polarity with Nile Red Solvatochromic Dye

Treated distillate aromatic extract (T-DAE) and mild extract solvate (MES) represent the most-used petroleum-based plasticizers in rubber compounds since 2010 when the use of distillate aromatic extract (DAE) was banned. T-DAE has an aromatic content in the range of 20–25%, while the aromatic content of MES is limited to 10–15%. Unfortunately, T-DAE is too dark to measure its E(NR) value but it was possible to perform a measurement on it with the $E_T(33)$ dye. On the other hand, MES is much more clear in color than T-DAE and it was possible to make a measurement both with $E_T(33)$ and Nile Red dyes as summarized in Table 1. From these data, T-DAE is appropriately positioned in Table 1 just between the S-SBR rubber and the polystyrene polymer. Surprisingly, the MES, which is prevalently aliphatic and naphthenic in its chemical nature, is positioned not among the hydrocarbons at the top of Table 1 but rather in the upper-mid part of Table 1, sharing the same E(NR) value as biodiesel i.e., the methyl ester of fatty acids from rapeseed oil, which instead is appropriately positioned in Table 1. There are probably components and impurities in the MES (which is a mixture of hydrocarbons) that affect its real polarity value. The presence of water and heteroatoms (e.g., N, S, Cl) may significantly alter the polarity in these industrial compounds.

As already stated in Section 2.1, there is a trend to use vegetable oils as rubber plasticizers, and the most popular are soybean and sunflower oils [19–21]. In Table 1, both oils were tested with the Nile Red dye, and the soybean oil appears to be slightly less polar with an E(NR) = 54.55 kcal/mol with respect to the sunflower oil (high oleic content), which has an E(NR) = 54.13 kcal/mol. It is also interesting that a commercial product Nytex Bio 6200, which is a blend of petroleum-based naphthenic oil and fatty acids, is classified by the E(NR) scale to be just in the middle between soybean oil and sunflower oil.

On the other hand, the methyl esters of fatty acids derived from rapeseed oil (biodiesel) or from coconut oil are slightly less polar than triglycerides. In fact, for the methyl esters of fatty acids, E(NR) = 54.7 kcal/mol, while the triglycerides show E(NR) = 54.1–54.5 kcal/mol. Thus, at least in terms of compatibility with rubbers for tire application, based on the above data, there are no practical differences between the methyl esters of fatty acids and the triglycerides, as already disclosed some time ago [17].

In the past, phthalates were used as plasticizers for winter tire treads. However, at present, there are a number of concerns regarding the environmental and health impact of phthalates [45–47] so they must be replaced by safer plasticizers. The typical plasticizers used in place of phthalates are adipates and sebacates. In our E(NR) scale, it is possible to see that the phthalates are located at E(NR) = 55.0–55.3 kcal/mol, although DOP (diethylhexylphthalate), the most used in the past, was found at 53.2 kcal/mol. It is interesting to note that the adipates are found at about 55 kcal/mol, with diethylhexyladipate (DOA) being the most common at 54.67 kcal/mol. The same comments apply to sebacates, which, although more expensive than adipates, display similar E(NR) values to adipates and DOA in particular.

Another trend regarding the substitution of DOP involves the use of terephthalic acid derivatives, and in particular, the use of dioctylterephthalate (DOTP), whose E(NR) in Table 1 was found to be 54.27 kcal/mol, which was less polar, as expected, than DOP at 53.24 kcal/mol.

Tetrahydrofurfuryl alcohol (THFA) has been proposed as an ideal alcohol from renewable sources. In fact, it derives from furfurol obtained from mineral acid treatment of biomasses. Furfurol is fully hydrogenated to THFA. We have synthesized several esters of THFA as part of another project [48] and in Table 1, we report the polarity values of the adipate, sebacate, oleate, laurate and pelargonate measured with the $E_T(33)$ dye; in the Discussion section, it will be shown how the $E_T(30)$ values correlate linearly with the E(NR) values for the class of compounds considered in the present paper. The THFA esters appear too polar to have good compatibility with common rubbers such as IR, BR and S-SBR but may be more than suitable as plasticizers for nitrile rubber (NBR) with E(NR) = 53.07 kcal/mol and 52-53 kcal/mol for the THFA esters.

Polyethylene glycol is used in certain rubber compounds as a compatibilizer aid between silica filler and rubber. The E(NR) value for PEG-400 was found at 52.15 kcal/mol, while the E(NR) value of raw silica surfaces is not known, but a work on modified silica surfaces reports E(NR) ≈ 48-7–52.6 kcal/mol (depending from the type and degree of modification of the surface) [49]. Based on these data, PEG-400 is really suitable for interacting with silica surfaces. In Table 1, it is shown that polytetrahydrofuran (polyTHF) oligomer is also a suitable liquid for the compatibilization of silica surfaces with rubber, since polyTHF shows the same E(NR) value as PEG-400. To reduce the polarity of PEG-400, it is possible to esterificate the OH end groups of the glycol with oleic acid to obtain either PEG monooleate with E(NR) = 53.6 kcal/mol or PEG dioleate with 53.8 kcal/mol.

## 4. Discussion

### 4.1. Correlation between the E(NR) Scale and the $E_T(30)$ Scale

The results have shown that Nile Red is an excellent and complementary solvatochromic dye with respect to $E_T(30)$. It is complementary because it is soluble in hydrocarbons and it is not sensitive to the residual acidity that may be present in ester plasticizers. Furthermore, Nile Red is also soluble in the same solvents as $E_T(30)$ and $E_T(33)$. Without the use of Nile Red, the present work could not have been conducted since we dealt with hydrocarbon rubber polymers and plasticizers where $E_T(30)$ was insoluble or problematic but where $E_T(33)$ also had problems. Furthermore, $E_T(33)$ is no longer easily commercially available, while the easy availability of Nile Red is due to the fact that it is also used as a staining dye for biological tissues and cells [27].

In this work, 53 compounds, either polymers or plasticizers, were studied with Nile Red and wherever possible also with the $E_T(33)$ dye. The data in Table 1 obtained with the latter dye were converted to the $E_T(30)$ scale using Equation (15). Only in a couple of cases was it possible to use $E_T(30)$ dye directly. The experimental data in Table 1 were then put in a graph as shown in Figure 2, so that the following equation was obtained with a very good correlation coefficient $R^2 = 0.944$:

$$E(NR) = -0.3039 \, E_T(30) + 67.312 \tag{17}$$

and the reverse equation,

$$E_T(30) = [E(NR) - 67.312]/(-0.3039) \qquad (18)$$

Equation (17) was then used to calculate the E(NR) values which were not measured with Nile Red dye. The calculated values are shown in italic characters in Table 1. Similarly, for the $E_T(30)$ values in Table 1, use was made of Equation (18), and the calculated values are shown in italic characters as well.

### 4.2. Correlation between the Solubility Parameter $\delta_t$ and the E(NR) Scale

In the introduction, we presented the classical approach to determine the compatibility between a polymer and a plasticizer. The classical approach involves the determination of the solubility parameter of both components, by finding these values in published tables (e.g., Hansen's book [13]) or calculating the solubility parameter of each component by group increments through the Van Krevelen method [14] or other similar approaches. Then, the solubility criteria are shown in Equations (11)–(13).

It is interesting at this point to evaluate how the total solubility parameter $\delta_t$ correlates with the E(NR) scale. Our first approach was to use the Hansen solubility parameter $\delta_t$ derived from the data tabulated in ref. [13]. However, the correlation with the E(NR) values reported in Table 1 was not satisfactory.

The following step was to calculate the $\delta_t$ value for each solvent, plasticizer and polymer reported in Table 1 according to the Van Krevelen method [14]. The calculation results are shown in the last column on the right of Table 1. These calculated $\delta_t$ values were then put in a graph against the E(NR) values of Table 1 and the results are shown in Figure 3. Although this time, the correlation coefficient $R^2 = 0.64$ is not as good as the E(NR) vs ET(30) correlation shown in Figure 1, an evident trend can be observed so that at least a rough estimation of E(NR) can be derived from $\delta_t$ according to the following:

$$\delta_t = -0.472\,E(NR) + 43.33 \qquad (19)$$

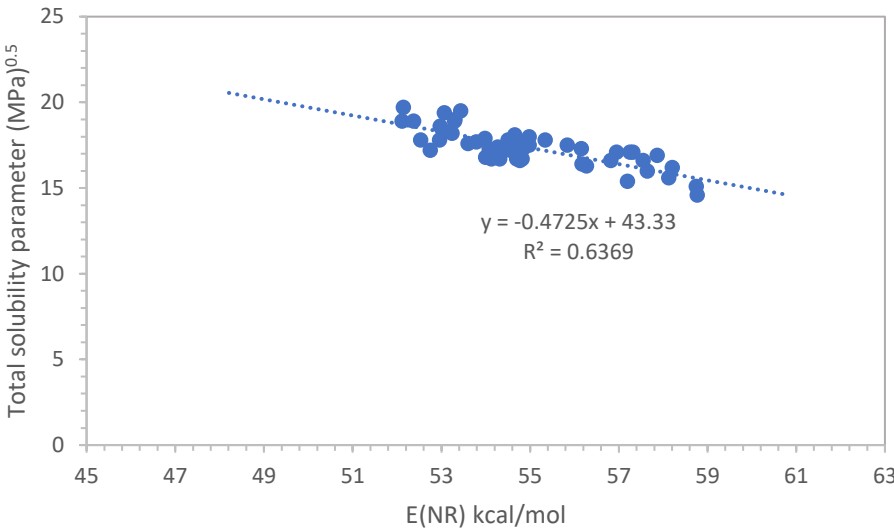

**Figure 3.** Correlation between the solubility parameter $\delta_t$ and the E(NR) scale.

In the correlation between the $\delta_t$ values of Table 1 and the E(NR) values, we were forced to exclude the $\delta_t$ values of tetrahydrofurfuryl alcohol, ethyl lactate and the polymer polymethylmethacrylate (PMMA).

### 4.3. Some Reflections on the Diene Rubber Compatibility Based on the E(NR) Scale

The main purpose of the present work was to create a polarity scale for hydrocarbon rubbers and conventional and new plasticizers. As shown in Table 1, it was possible

to measure the E(NR) value of the key diene rubbers polyisoprene (IR), polybutadiene (BR) and the copolymer styrene–butadiene (S-SBR). From this measurement, it turned out that the liquid oligomers of IR and BR were adequate model compounds of the higher polymers IR and BR since very similar E(NR) values were measured. The same reasoning also applies for squalene as a model compound of IR polymer. Regarding the conventional hydrocarbon plasticizers of diene rubber, from Table 1, it is immediately evident that the T-DAE plasticizer with its relatively high aromatic content is fully compatible with S-SBR, since both compounds share the same E(NR) value. Furthermore, it is also confirmed that T-DAE is also more compatible with IR and BR than MES, which, although less aromatic, evidently contains some structural feature that increases its polarity to the same level as biodiesel (the methyl ester of rapeseed oil tested previously with success as a rubber compound plasticizer [17]).

The phthalate-based plasticizers used in the past in passenger winter tire treads result in the E(NR) scale being less polar and more compatible with the diene rubber than the currently used ester plasticizers like adipates and sebacates.

"Green" plasticizers are those produced from renewable sources. Thus, emerging green plasticizers for the tire industry are soybean and sunflower oils as well as the semi-green blend of naphthenic oil with fatty acids (Nytex Bio 6200). As expected, these green plasticizers are in the middle of Table 1, and the compatibility with the diene rubbers like IR, BR and S-SBR is probably moderate. On the other hand, in Table 1, it is immediately evident that such mentioned green plasticizers have excellent compatibility with more polar rubbers like epoxidized natural rubber (ENR) and nitrile rubber (NBR).

Tetrahydrofurfuryl alcohol (THFA) is considered a green alcohol fully derived from renewable sources. Thus, its esters with carboxylic acids also from renewable sources are 100% green. However, such esters of THFA are all found at the bottom of Table 1, suggesting high polarity and poor compatibility with IR, BR and S-SBR. Instead, the THFA esters are certainly suitable as plasticizers for ENR and NBR.

## 5. Conclusions

The evaluation of the polymer–plasticizer compatibility through an $E_T(30)$ scale is hindered by the insolubility of the $E_T(30)$ dye in hydrocarbon polymers like IR, BR and S-SBR. Furthermore, $E_T(30)$ dye may have limited solubility in plasticizers and is sensitive to their weak acidity. In fact, protonation of the $E_T(30)$ dye destroys the CT band of this solvatochromic dye. To circumvent such a situation, other authors [25] have proposed an interesting, original and sophisticated approach that permits the estimation of the $E_T(30)$ values of certain "difficult" substrates where the direct polarity measurement is hindered. The alternative approach proposed in the present work is to use Nile Red dye as a solvatochromic dye. Nile Red has the advantage of being soluble in hydrocarbons and in polar solvents and it is not sensitive to protonation.

In Table 1, a series of 53 different compounds including rubbers, plasticizers, hydrocarbon solvents and even some polar polymers (like PMMA, PLLA, PEG-400 and polyTHF) were tested with the Nile Red probe. For the evaluation of liquid polarity, Nile Red was dissolved in the selected liquid and the spectrum was recorded, while for polymers, Nile Red was embedded in a thin solid film of the selected polymer and the spectrum was recorded on the solid-state film. Wherever possible, the liquid or plasticizer was also evaluated with the $E_T(33)$ dye (seldom with the $E_T(30)$ dye) and the resulting value converted into the $E_T(30)$ scale through Equation (15).

Thus, Table 1, in a certain number of cases, reports both the E(NR) value and the $E_T(30)$ value. These data were used to derive a correlation between E(NR) scale and $E_T(30)$, limited to a selected number of polymers and plasticizers, as shown in Figure 2 and displayed by Equations (17) and (18) with a very good correlation coefficient.

Furthermore, the total solubility parameter $\delta_t$ as defined by Equation (10) has also been calculated for each compound according to the Van Krevelen method [14] and reported in

Table 1. A reasonable correlation was also found between the solubility parameter and the E(NR) scale, as shown in Figure 3 and described by Equation (19).

Thus, with Nile Red dye, it is possible to study the polarity of hydrocarbon polymers (like rubbers) and plasticizers and to connect the results either with Reichardt's $E_T(30)$ scale or with the total solubility parameter.

**Funding:** This research received no external funding.

**Data Availability Statement:** This work will be available on the Researchgate website on the author's page.

**Conflicts of Interest:** The author declares no conflicts of interest.

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
