# Peer review of "Conventional and Green Rubber Plasticizers Classified through Nile Red [E(NR)] and Reichardt’s Polarity Scale [ET(30)]"

_liquids, doi:10.3390/liquids4020015_

Round 1
Reviewer 1 Report
Comments and Suggestions for Authors
This is an interesting paper, which is important for understanding of the polarity of plasticizers. The results are well documented and are valuable for applied science and technology. This work should be published.
The most important question is as follows. As commercial products of such type may contain stabilizers, how can the last as well as (possible) impurities and dissolved oxygen and carbon dioxide affect the results obtained with indicators?
Another problem consists of the possibility of the formation of dye microcrystals in the evaporated sample. Such effects may appear on the adsorption of dyes on solids (see the enclosed file, page 2, Left column).
Minor correction: Page 6, Line 237: instead of “four orders of magnitude” it will be better to say about “four logarithmic units”.

Author Response
THE ANSWER TO THE REFEREES ARE REPORTED IN CAPITAL BOLD AND YELLOW SHADED
Referee 1
Comments and Suggestions for Authors
This is an interesting paper, which is important for understanding of the polarity of plasticizers. The results are well documented and are valuable for applied science and technology. This work should be published.
THANK YOU FOR THE KIND COMMENTS.
The most important question is as follows. As commercial products of such type may contain stabilizers, how can the last as well as (possible) impurities and dissolved oxygen and carbon dioxide affect the results obtained with indicators?
THIS IS CERTAINLY AN IMPORTANT POINT. REGARDING THE STABILIZERS, IN GENERAL THEY ARE NOT PRESENT NEITHER IN THE PETROLEUM-BASED PLASTICIZERS NOT IN THE PLASTICIZERS PURCHASED FROM MERCK-ALDRICH.
THE STABILIZERS ARE INSTEAD PRESENT IN ALL POLYMERS BUT USUALLY THEIR CONCENTRATION IS ABOUT 0.1% THE MASS OF THE POLYMER. THUS, IT IS ASSUMED THAT THEIR POTENTIAL EFFECT ON THE SOLVATOCHROMIC DYE IS NEGLIGIBLE. REGARDING OTHER POSSIBLE IMPURITIES IN THE PLASTICIZERS, NO MEASURES WERE TAKE TO REMOVE THEM. IT WAS HOWEVER ADOPTED A STRICT BACKCORRECTION PROCEDURE AT THE SPECTROPHOTOMETER (AS DETAILED IN THE EXPERIMENTAL SECTION OF THE PAPER) TO MITIGATE THE EFFECT OF SUCH IMPURITIES, ASSUMING ONCE AGAIN THAT THE INTERFERENCE WITH THE SOLVATOCHROMIC DYE IS NEGLIGIBLE.
Another problem consists of the possibility of the formation of dye microcrystals in the evaporated sample. Such effects may appear on the adsorption of dyes on solids (see the enclosed file, page 2, Left column).
THIS IS ANOTHER POINT OF ABSOLUTE IMPORTANCE WHEN MEASURING THE MICROPOLARITY OF POLYMER FILMS. FIRST OF ALL, THE PAPER INDICATED BY THE REFEREE HAS BEEN INCLUDED IN THE REFERENCES OF THE REVISED VERSION OF THE PAPER (SEE REF. [22]). FURTHERMORE, IN THE EXPERIMENTAL SECTION IT WAS ADDED A SMALL PARAGRAPH AS FOLLOWS:
“IT IS KNOWN FROM LITERATURE [22] THAT THE SOLVATOCHROMIC DYE MAY SEPARATE UNDER THE FORM OF MICROCRYSTALS ON THE POLYMER SURFACE LEADING TO NOT USEFUL ABSORPTION MAXIMA READINGS. THIS POTENTIALLY UNDESIRED EFFECT WAS MINIMIZED BY USING THE MINIMUM POSSIBLE AMOUNT OF DYE DURING THE PREPARATION OF THE COMPOSITE. FURTHERMORE, NILE RED DYE IS CHARACTERIZED BY A HIGH SOLUBILITY IN HYDROCARBON POLYMERS (IN CONTRAST TO THE ET(30) AND ET(33) DYES). CONSEQUENTLY, THE POTENTIAL BLOOMING IS THEREFORE COMPLETELY SUPPRESSED.”
Minor correction: Page 6, Line 237: instead of “four orders of magnitude” it will be better to say about “four logarithmic units”.
WE HAVE PREFERRED TO KEEP “ORDERS OF MAGNITUDE”
Reviewer 2 Report
Comments and Suggestions for Authors
The author of this study tested the solvatochromic dye Nile red as a colorimetric probe to characterize the polarity of rubber polymers, copolymers and plasticizers to evaluate their polymer-plasticizer compatibility criteria. Unlike the traditional polarity indicator Reichardt’s dye, Nile red is soluble in all these materials and is insensitive to acidity. It was shown that the solvent polarity parameter measured with Nile red, E(NR), correlates with the polarity parameter measured with Reichardt’s dye, ET(30), and also with the total solubility parameter proposed by Van Krevelen. These are valuable results and the paper is written thoroughly, therefore I recommend it for publication in Liquids, noting that the manuscript requires a minor correction.
I ask the author to consider to mention that
- as can be seen in Figs. 1 and 2 – the absorption maxima of Nile blue and Reichardt’s dye vary in the opposite direction with the polarity of the medium, i. e. Nile blue is a positively solvatochromic dye, whereas Reichardt’s dye is negatively solvatochromic.
- as a disadvantage of the E(NR) scale – the position of the absorption band of NR is less sensitive to the polarity than the position of the band of Reichardt’s dye. This can also be seen in Figs. 1 and 2.
Further comments
- row 11: and „is not also useful” correctly „is also not useful”
- row 39. “tents” correctly “tens”
- row 45: “n the moles of plasticizer added” is n the notation of the molar fraction of the plactisizer added?
- rows 45 and 48: if “k” mean different proportionality constants, different notations should be used
- row 82: “chains, ends and side groups movements” correctly “chain, end and side group movements”
- row 164: delete “Either”
- row 345: “there is trend” correctly “there is a trend”
- row 439: “mail purpose” correctly “main purpose”
Comments on the Quality of English LanguageCorrect English
Author Response
THE ANSWER TO THE REFEREES ARE REPORTED IN CAPITAL BOLD AND YELLOW SHADED
Referee 2
The author of this study tested the solvatochromic dye Nile red as a colorimetric probe to characterize the polarity of rubber polymers, copolymers and plasticizers to evaluate their polymer-plasticizer compatibility criteria. Unlike the traditional polarity indicator Reichardt’s dye, Nile red is soluble in all these materials and is insensitive to acidity. It was shown that the solvent polarity parameter measured with Nile red, E(NR), correlates with the polarity parameter measured with Reichardt’s dye, ET(30), and also with the total solubility parameter proposed by Van Krevelen. These are valuable results and the paper is written thoroughly, therefore I recommend it for publication in Liquids, noting that the manuscript requires a minor correction.
I ask the author to consider to mention that
- as can be seen in Figs. 1 and 2 – the absorption maxima of Nile Red and Reichardt’s dye vary in the opposite direction with the polarity of the medium, i. e. Nile Red is a positively solvatochromic dye, whereas Reichardt’s dye is negatively solvatochromic.
- as a disadvantage of the E(NR) scale – the position of the absorption band of NR is less sensitive to the polarity than the position of the band of Reichardt’s dye. This can also be seen in Figs. 1 and 2.
THE TWO POINTS ABOVE WERE MENTIONED IN THE REVISED VERSION OF THE PAPER.
Further comments
MANY THANKS FOR THE CORRECTIONS: ALL THE SUGGESTED CORRECTIONS WERE IMPLEMENTED IN THE REVISED MANUSCRIPT
- row 11: and „is not also useful” correctly „is also not useful”
- row 39. “tents” correctly “tens”
- row 45: “n the moles of plasticizer added” is n the notation of the molar fraction of the plactisizer added?
n IS NUMBER OF MOLES OF PLASTICIZER ADDED, NOT THE MOLAR FRACTION
- rows 45 and 48: if “k” mean different proportionality constants, different notations should be used
- row 82: “chains, ends and side groups movements” correctly “chain, end and side group movements”
“POLYMER IS A MEASURE OF THE INTERNAL AVAILABLE SPACE FOR THE MOTION OF THE CHAIN SEGMENTS, THE CHAIN ENDS AND THE SIDE GROUPS MOTION”
- row 164: delete “Either”
- row 345: “there is trend” correctly “there is a trend”
- row 439: “mail purpose” correctly “main purpose”
Comments on the Quality of English Language
Correct English
THE ENGLISH LANGUAGE WAS CAREFULLY CHECKED AND IMPROVED WHEREVER NECESSARY
Reviewer 3 Report
Comments and Suggestions for Authors
Liquids-2846609
Reviewer’s Comments
In the paper, “Conventional and green Rubber Plasticizers classified through the Nile Red [E(NR)] and Reichardt’s Polarity Scale [ET(30)],” Cataldo presents an alternative or complementary approach to the solubility parameter to assess the compatibility between a plasticizer and a polymer matrix by classifying the plasticizers and polymers through the Reichardt’s ET(30) polarity scale, as well as a complementary scale based on the Nile Red dye.
The paper is fairly well-written, but I did find several minor grammatical and typographical errors. While these errors were not detrimental to the flow of the paper or the presentation of the results, I encourage the authors to carefully comb the paper and fix any glaring errors. Below I’ve included a list of some the errors that I found.
1. Abstract, p. 1 (lines 19-20): Change “…conventional plasticizers as well as some new and green plasticizer proposed for the rubber compounds.” to “…conventional plasticizers, as well as some new and green plasticizers proposed for the rubber compounds.”
2. Introduction, p. 2 (lines 83-87): I suggest numbering the list. “The increase of the motion of these moieties of the polymer can be achieved by (1) heating, (2) the addition of a plasticizer, which being a low molecular weight molecule increases the number chain ends dramatically, (3) introducing branching or bulky side groups to the main polymer chain, and (4) inserting more flexible chain segments into the main polymer chain.”
3. Results, p. 8 (line 283-284): The sentence, “The Nile Red dye is steady employed in many measurements ranging probing hydrocarbon liquids…” is a bit confusing. Should this read something like, “The Nile Red dye is regularly employed in many measurements ranging from probing hydrocarbon liquids…” Should this Also, as in the previous example, a numbered list may be more suitable here.
4. Results, p. 8 (line 294-295): Perhaps change “Dutta and col.” to “Dutta and co-workers.” or “Dutta et al.”
5. Results, p. 8 (line 298): Change “Furthermore, also the micropolarity of polystyrene and poly(lactic acid) films were studied, while for the PMMA it was adopted the E(NR) value found by Dutta et al. [41].” to “The micropolarity of polystyrene and poly(lactic acid) films were also studied, whereas the E(NR) value reported by Dutta et al. was adopted for PMMA. [41]”
6. Results, p. 8 (line 304-305): Perhaps change “Regarding the rubber and other polymer studied, the trend from apolar to polar compounds is absolutely logic starting with the liquid model compounds…” to “Regarding the rubber and other polymer studied, there is a logical trend from apolar to polar compounds starting with the liquid model compounds…”
7. Discussion, p.12 (line 421): Perhaps change “…shown in the eq. (11-13)…” to “…shown in eq. (11) - (13)…”
8. Discussion, p.13 (line 439): Change “The mail purpose…” to “The main purpose…”
9. Discussion, p.14 (line 465-466): Change “…suggesting an high level of polarity…” to “…suggesting high polarity…”
10. Conclusions, p.14 (lines 470-471): Change “…hydrocarbon polymers like for instance IR, BR and S-SBR.” to “…hydrocarbon polymers like IR, BR, and S-SBR.”
11. Conclusions, p.14 (lines 472): Change “In fact protonation of the ET(30) dye…” to “In fact, protonation of the ET(30) dye…”.
12. Conclusions, p.14 (lines 488): Change “This has permitted to derive a correlation between…” to “This data was used to derive a correlation between…”
The purpose and scope of the research is straightforward, and this made for a succinct and easy to follow paper. The author does a good job explaining the motivation of the research and providing appropriate background information, and the title, abstract, figures and figure captions, and references are all appropriate. The results and conclusions are well-presented, and the interpretation of the results and conclusions are sound and well-supported by both the data and the available literature.
In my opinion, this paper will be of interest to the readership of Liquids, and I recommend publication after minor revisions.

The paper is fairly well-written, but I did find several minor grammatical and typographical errors. While these errors were not detrimental to the flow of the paper or the presentation of the results, I encourage the authors to carefully comb the paper and fix any glaring errors. Below I’ve included a list of some the errors that I found.
1. Abstract, p. 1 (lines 19-20): Change “…conventional plasticizers as well as some new and green plasticizer proposed for the rubber compounds.” to “…conventional plasticizers, as well as some new and green plasticizers proposed for the rubber compounds.”
2. Introduction, p. 2 (lines 83-87): I suggest numbering the list. “The increase of the motion of these moieties of the polymer can be achieved by (1) heating, (2) the addition of a plasticizer, which being a low molecular weight molecule increases the number chain ends dramatically, (3) introducing branching or bulky side groups to the main polymer chain, and (4) inserting more flexible chain segments into the main polymer chain.”
3. Results, p. 8 (line 283-284): The sentence, “The Nile Red dye is steady employed in many measurements ranging probing hydrocarbon liquids…” is a bit confusing. Should this read something like, “The Nile Red dye is regularly employed in many measurements ranging from probing hydrocarbon liquids…” Should this Also, as in the previous example, a numbered list may be more suitable here.
4. Results, p. 8 (line 294-295): Perhaps change “Dutta and col.” to “Dutta and co-workers.” or “Dutta et al.”
5. Results, p. 8 (line 298): Change “Furthermore, also the micropolarity of polystyrene and poly(lactic acid) films were studied, while for the PMMA it was adopted the E(NR) value found by Dutta et al. [41].” to “The micropolarity of polystyrene and poly(lactic acid) films were also studied, whereas the E(NR) value reported by Dutta et al. was adopted for PMMA. [41]”
6. Results, p. 8 (line 304-305): Perhaps change “Regarding the rubber and other polymer studied, the trend from apolar to polar compounds is absolutely logic starting with the liquid model compounds…” to “Regarding the rubber and other polymer studied, there is a logical trend from apolar to polar compounds starting with the liquid model compounds…”
7. Discussion, p.12 (line 421): Perhaps change “…shown in the eq. (11-13)…” to “…shown in eq. (11) - (13)…”
8. Discussion, p.13 (line 439): Change “The mail purpose…” to “The main purpose…”
9. Discussion, p.14 (line 465-466): Change “…suggesting an high level of polarity…” to “…suggesting high polarity…”
10. Conclusions, p.14 (lines 470-471): Change “…hydrocarbon polymers like for instance IR, BR and S-SBR.” to “…hydrocarbon polymers like IR, BR, and S-SBR.”
11. Conclusions, p.14 (lines 472): Change “In fact protonation of the ET(30) dye…” to “In fact, protonation of the ET(30) dye…”.
12. Conclusions, p.14 (lines 488): Change “This has permitted to derive a correlation between…” to “This data was used to derive a correlation between…”
Author Response
ANSWERS TO THE REVIEWER ARE IN CAPITAL BOLD CHARACTERS
Referee 3
Comments and Suggestions for Authors
Liquids-2846609
Reviewer’s Comments
In the paper, “Conventional and green Rubber Plasticizers classified through the Nile Red [E(NR)] and Reichardt’s Polarity Scale [ET(30)],” Cataldo presents an alternative or complementary approach to the solubility parameter to assess the compatibility between a plasticizer and a polymer matrix by classifying the plasticizers and polymers through the Reichardt’s ET(30) polarity scale, as well as a complementary scale based on the Nile Red dye.
The paper is fairly well-written, but I did find several minor grammatical and typographical errors. While these errors were not detrimental to the flow of the paper or the presentation of the results, I encourage the authors to carefully comb the paper and fix any glaring errors. Below I’ve included a list of some the errors that I found.
THANK YOU FOR THE KIND AND EXTREMELY USEFUL CORRECTION LIST. ALL THE SUGGESTED CORRECTIONS HAVE BEEN IMPLEMENTED IN THE REVISED VERSION OF THE PAPER.
- Abstract, p. 1 (lines 19-20): Change “…conventional plasticizers as well as some new and green plasticizer proposed for the rubber compounds.” to “…conventional plasticizers, as well as some new and green plasticizers proposed for the rubber compounds.”
- Introduction, p. 2 (lines 83-87): I suggest numbering the list. “The increase of the motion of these moieties of the polymer can be achieved by (1) heating, (2) the addition of a plasticizer, which being a low molecular weight molecule increases the number chain ends dramatically, (3) introducing branching or bulky side groups to the main polymer chain, and (4) inserting more flexible chain segments into the main polymer chain.”
- Results, p. 8 (line 283-284): The sentence, “The Nile Red dye is steady employed in many measurements ranging probing hydrocarbon liquids…” is a bit confusing. Should this read something like, “The Nile Red dye is regularly employed in many measurements ranging from probing hydrocarbon liquids…” Should this Also, as in the previous example, a numbered list may be more suitable here.
- Results, p. 8 (line 294-295): Perhaps change “Dutta and col.” to “Dutta and co-workers.” or “Dutta et al.”
- Results, p. 8 (line 298): Change “Furthermore, also the micropolarity of polystyrene and poly(lactic acid) films were studied, while for the PMMA it was adopted the E(NR) value found by Dutta et al. [41].” to “The micropolarity of polystyrene and poly(lactic acid) films were also studied, whereas the E(NR) value reported by Dutta et al. was adopted for PMMA. [41]”
- Results, p. 8 (line 304-305): Perhaps change “Regarding the rubber and other polymer studied, the trend from apolar to polar compounds is absolutely logic starting with the liquid model compounds…” to “Regarding the rubber and other polymer studied, there is a logical trend from apolar to polar compounds starting with the liquid model compounds…”
- Discussion, p.12 (line 421): Perhaps change “…shown in the eq. (11-13)…” to “…shown in eq. (11) - (13)…”
- Discussion, p.13 (line 439): Change “The mail purpose…” to “The main purpose…”
- Discussion, p.14 (line 465-466): Change “…suggesting an high level of polarity…” to “…suggesting high polarity…”
- Conclusions, p.14 (lines 470-471): Change “…hydrocarbon polymers like for instance IR, BR and S-SBR.” to “…hydrocarbon polymers like IR, BR, and S-SBR.”
- Conclusions, p.14 (lines 472): Change “In fact protonation of the ET(30) dye…” to “In fact, protonation of the ET(30) dye…”.
- Conclusions, p.14 (lines 488): Change “This has permitted to derive a correlation between…” to “This data was used to derive a correlation between…”